# HIV Increases the Inhibitory Impact of Morphine and Antiretrovirals on Autophagy in Primary Human Macrophages: Contributions to Neuropathogenesis

**DOI:** 10.3390/cells10092183

**Published:** 2021-08-24

**Authors:** John M. Barbaro, Ana Maria Cuervo, Joan W. Berman

**Affiliations:** 1Montefiore Medical Center, Department of Pathology, Albert Einstein College of Medicine, 1300 Morris Park Ave, Bronx, NY 10461, USA; john.barbaro@einsteinmed.org; 2Montefiore Medical Center, Department of Developmental and Molecular Biology, Albert Einstein College of Medicine, 1300 Morris Park Ave, Bronx, NY 10461, USA; ana-maria.cuervo@einsteinmed.org; 3Montefiore Medical Center, Department of Microbiology and Immunology, Albert Einstein College of Medicine, 1300 Morris Park Ave, Bronx, NY 10461, USA

**Keywords:** myeloid cells, HIV-associated neurocognitive disorders, opioids, antiretroviral therapy, autophagy, selective autophagy, LC3, p62/SQSTM1, mitophagy

## Abstract

HIV enters the CNS early after peripheral infection, establishing reservoirs in perivascular macrophages that contribute to development of HIV-associated neurocognitive disorders (HAND) in 15–40% of people with HIV (PWH) despite effective antiretroviral therapy (ART). Opioid use may contribute to dysregulated macrophage functions resulting in more severe neurocognitive symptoms in PWH taking opioids. Macroautophagy helps maintain quality control in long-lived cell types, such as macrophages, and has been shown to regulate, in part, some macrophage functions in the CNS that contribute to HAND. Using Western blotting and confocal immunofluorescence in primary human macrophages, we demonstrated that morphine and a commonly prescribed ART regimen induce bulk autophagy. Morphine and ART also inhibited completion of autophagy. HIV infection increased these inhibitory effects. We also examined two types of selective autophagy that degrade aggregated proteins (aggrephagy) and dysfunctional mitochondria (mitophagy). Morphine and ART inhibited selective autophagy mediated by p62 regardless of HIV infection, and morphine inhibited mitophagic flux in HIV-infected cells demonstrating potential mitotoxicity. These results indicate that inhibition of autophagy, both in bulk and selective, in CNS macrophages may mediate neurocognitive dysfunction in PWH using opioids. Increasing autophagic activity in the context of HIV may represent a novel therapeutic strategy for reducing HAND in these individuals.

## 1. Introduction

Approximately 38 million people worldwide are living with HIV [1]. Antiretroviral therapy (ART) has dramatically improved both the length and quality of life for people with HIV (PWH), yet comorbidities persist that can increase morbidity and mortality [2]. One significant comorbidity is HIV-associated neurocognitive disorders (HAND), a spectrum of neurological illnesses that affect about 15–40% of PWH despite virally suppressive ART [3,4]. The mechanisms by which HAND occurs in the ART era are important to determine such that effective therapies can be developed to reduce this burdensome comorbidity. Our laboratory and others have shown that HIV enters the CNS by a subpopulation of mature monocytes expressing CD14, the LPS coreceptor, and CD16, the FcγIII receptor [5]. These monocytes are preferentially infected with HIV and cross the blood–brain barrier (BBB) more than uninfected cells, seeding the CNS with HIV within 1–2 weeks after peripheral infection [6,7]. Once in the CNS, these monocytes can differentiate into long-lived perivascular macrophages that can infect other macrophages, microglia, and, to a lesser extent, astrocytes [8,9]. Cell functions of macrophages are dysregulated in the context of HIV to perpetuate neuroinflammation that contributes to neuronal damage and loss and, thus, HAND development. These include aberrant phagocytosis that diminishes clearance of neurotoxic debris, increased reactive oxygen and nitrogen species (ROS/RNS) production that activates other cells, and increased inflammatory cytokines that injure neurons directly and recruit more uninfected and infected immune cells into the CNS [10,11,12].

Substance use is closely linked to the HIV epidemic, which increases the possibility of HIV transmission [13,14]. Some studies have also shown that substance use increases neurocognitive dysfunction characteristics of HAND in PWH, including in those using opioids [15,16,17,18]. Approximately 10 million Americans misused prescription opioids in 2019, and over 700,000 used heroin, which is often taken intravenously and is then metabolized to morphine [19]. Preliminary data also indicate that opioid use has increased during the COVID-19 pandemic [20]. A more detailed understanding of how opioids impact mechanisms of HIV neuropathogenesis will guide development of interventional strategies to treat people with HAND who use opioids. Macrophages in the CNS are cellular reservoirs for HIV and express all three subtypes of the opioid receptors, μ, κ, and δ [9,21,22,23]. Thus, opioids can alter macrophage functions to contribute to increased CNS damage. Some studies also indicate that specific antiretroviral drugs are neurotoxic and can promote inflammation in certain cell types by increasing cytokine expression and ROS levels [24,25,26,27]. This underscores the need to characterize the impact of opioids on macrophage functions in the context of both HIV and ART.

Dysregulation of macrophage functions in response to HIV, ART, and opioids is likely related to disruption of important, homeostatic cell processes. One such process is autophagy, a series of quality control mechanisms that degrade damaged and dysfunctional cytosolic macromolecules in the endolysosomal system [28]. The most well-studied type is macroautophagy, hereby termed autophagy, which can degrade cargo “in bulk” or selectively target specific cellular components for degradation [29]. In autophagy, a double-membraned organelle, the autophagosome (APG), forms and encloses around cytosolic cargo to be degraded [30]. During this process, Microtubule-associated Protein 1A/1B-Light Chain 3 (LC3) gets lipidated, forming LC3II, and associates with the inner and outer membrane of the forming APG, serving as a marker to monitor autophagic activity [31]. While most APG cargo is sequestered in bulk, cargo receptors and adaptors, such as Sequestrome-1 (p62), can selectively target specific types of cargo to APG [29]. These receptors/adaptors bind distinct molecules in the cargo and dock them to the forming APG by interacting with LC3II on the inner membrane of this organelle [29,32]. Cargo docked by p62 includes aggregated proteins with polyubiquitination at K63. One of the most studied types of selective autophagy is mitophagy, whereby autophagy receptors anchor damaged mitochondria for sequestration into APG (mitophagosomes) and subsequent degradation in lysosomes. [29,32]. Once the APG forms, signaling events activate SNARE proteins, scaffold tethering proteins, and Rab family GTPases to promote fusion of the APG outer membrane with late endosomes/lysosomes in a process termed maturation. APG maturation leads to formation of autolysosomes (AL), single-membraned organelles where cargo is broken down by lysosomal enzymes [33]. Building blocks of the digested materials are then recycled to sustain cellular anabolic processes. The total process of APG biogenesis to digestion of cargo in AL is termed autophagic flux [31]. HIV is known to induce autophagy in macrophages and may impair APG maturation to inhibit flux [34,35]. Studies of the impact of opioids, such as morphine, or ART drugs on autophagy in human macrophages are lacking, particularly in the context of HIV. In this work, we used uninfected and HIV-infected primary human macrophages to determine the effect of morphine and/or a common ART regimen used to both treat and prevent HIV infection on in bulk autophagy and two types of selective autophagy, p62-dependent autophagy and mitophagy. We found that morphine and ART in combination have synergistic effects on autophagy and that these effects are different when macrophages are infected with HIV. Interestingly, the interaction of morphine, ART, and HIV is specifically detrimental for selective forms of autophagy such as p62-dependent autophagy and mitophagy. Reversing inhibited in bulk and selective autophagy in response to HIV, morphine, and ART may ameliorate functional dysregulation of macrophages to treat HAND in PWH who use opioids.

## 2. Materials and Methods

### 2.1. Cell Culturing, Treatments, and Infection with HIV

Leukopaks from deidentified individuals were obtained from the New York Blood Center. No demographic information other than blood type was known. Ficoll gradient centrifugation was used to isolate peripheral blood mononuclear cells (PBMC). For Western blotting, RT-qPCR, and p24 alphaLISA experiments, 15–40 million PBMC were seeded on 60 mm or 100 mm tissue culture-treated dishes in Dulbecco’s Modified Eagle Medium (DMEM) containing 10% FBS (Gibco), 5% human AB serum (Corning Inc., Corning, NY, USA), 1% penicillin/streptomycin (Gibco Technologies, Amarillo, TX, USA), 1% glutamine (Gibco Technologies, Amarillo, TX, USA), and 1% 1M HEPES with 10–20 ng/mL of macrophage colony-stimulating factor, M-CSF (Peprotech Cat. No.: 300-25, East Windsor, NJ, USA), at 37 °C with 5% CO_2_. After 3 days, media was changed, and cells were cultured with M-CSF for 3 additional days to differentiate into monocyte-derived macrophages (MDM). For microscopy experiments, monocytes were isolated by negative selection using the MojoSort Pan Monocyte Isolation Kit (Biolegend Cat. No.: 480060, San Diego, CA, USA), and 100,000 monocytes/well were plated on PDL-coated round glass coverslips (Corning) in 24-well plates. Uninfected MDM were untreated (Untx) or treated with 100 nM morphine (Sigma-Aldrich, St. Louis, MO, USA), an ART cocktail consisting of 15 μM tenofovir, 15 μM emtricitabine, and 1 μM raltegravir, or both morphine+ART, for 24 h. All three ART drugs were used in their free base forms. Tenofovir and emtricitabine concentrations are consistent with prior studies in primary human macrophages and monocytes [36,37,38,39]. The concentration of raltegravir used is consistent with studies that tested integrase inhibitors in myeloid cells, and it is within the range detectable in serum [40,41]. ART compounds were obtained through the NIH AIDS Reagent Program, Division of AIDS, NIAID, NIH, as follows: tenofovir (Cat. No.: 10199); emtricitabine (Cat. No.: 10071); and raltegravir (Cat. No.: 11680). Morphine and all ART drugs were dissolved in water. For experiments with infected cells, day 6 MDM were infected with 20 ng/mL HIV_ADA_ (NIH, Bethesda, MD, USA). After 24 h, media was changed, and MDM were cultured for 2–3 additional days prior to treatment with morphine with or without ART for 24 h.

### 2.2. HIV Quantification

A 1 mL volume of supernatants from confluent 60 mm dishes was collected to measure HIV p24 levels by a p24 alphaLISA kit (Perkin-Elmer, Waltham, MA, USA) after 4–5 days total of infection. Assays were performed according to the manufacturer’s protocol, with samples run in duplicate per condition. Values of HIV p24 in pg/mL were interpolated using a sigmoid standard curve. These values were then averaged per condition across experiments.

### 2.3. Autophagic Flux Measurement

Uninfected or HIV-infected MDM were treated with morphine and/or ART for 24 h or left untreated. To assess LC3II and p62 flux, lysosomal inhibitors, 10–20 mM NH_4_Cl + 100–200 μM leupeptin (Sigma-Aldrich, St. Louis, MO, USA) (N+L), were added to some dishes in the last 4 h of treatment [31]. Inhibitor concentrations were optimized by treating MDM with N+L in the presence and absence of vinblastine (Sigma-Aldrich, St. Louis, MO, USA), a microtubule polymerization inhibitor that impairs autophagosome fusion with lysosomes [42,43]. The optimal time and dose of N+L did not cause any additional accumulation of LC3II in the presence of vinblastine, indicating full blockage of LC3II degradation [44]. After treatments, plates were washed 3× with cold PBS and lysed with Radioimmunoprecipitation Assay (RIPA) buffer containing Halt Protease Inhibitor and Phosphatase Inhibitor Cocktail (Thermo Fisher Scientific, Waltham, MA, USA). Protein concentrations in each lysate were determined by the Bradford method using Protein Assay Reagent Concentrate (Bio-Rad, Hercules, CA, USA). A 20–50 μg amount of protein/lane was resolved by sodium dodecyl sulfate–polyacrylamide gel electrophoresis (SDS-PAGE) under reducing conditions, followed by transfer overnight at 4 °C on nitrocellulose membranes (GE Healthcare, Chicago, IL, USA). Revert Total Protein Stain (Li-Cor, Lincoln, NE, USA) was used to quantify total protein levels following transfer using the Odyssey Fc System (Li-Cor, Lincoln, NE, USA) for visualization in the dynamic range of signals. Image Studio v.5.2 software (Li-Cor, Lincoln, NE, USA) was used to quantify the optical density of all signals. The total protein loading control is essential for these studies, as specific proteins commonly used as loading controls (the so-called housekeeping proteins) have also been shown to be degraded by autophagy and could mask findings [43]. After total protein staining, membranes were blocked in 5% nonfat dry milk in 1xTBS with 0.1% Tween-20 prior to incubation overnight at 4 °C with rabbit anti-LC3B (52 ng/mL, 1:1000 Cell Signaling Technology #2775, Danvers, MA, USA) or with rabbit anti-p62 (0.5 μg/mL, 1:1000 Enzo Life Sciences BML-PW9860, Ann Arbor, MI, USA) antibodies. HRP-conjugated goat anti-rabbit (65.7 ng/mL, 1:1000 Cell Signaling Technology #7074, Danvers, MA, USA) was used as the secondary antibody for both LC3 and p62. Blots for LC3 and p62 were developed using 1:1 Super SignalWest Femto Chemiluminescent Substrate and Luminol/Enhancer (Thermo Fisher Scientific, Waltham, MA, USA) and quantified as for total protein. 

There is variability inherent in primary human cells. Thus, the amount of protein obtained from MDM derived from different donors can vary considerably [6,45]. To mitigate this, LC3II and p62 levels were normalized by two methods. Firstly, the signal for LC3II and p62 was normalized to total protein levels. Secondly, LC3II levels for two experiments in HIV-infected MDM were normalized to β-actin levels, demonstrating no significant difference from total protein normalization for these two sets of cells (Appendix A). To determine changes in different autophagy parameters, LC3II or p62 was analyzed in three ways. LC3II or p62 steady-state protein levels were determined as the amount of normalized protein in cells that did not receive N+L, and this normalized LC3II or p62 value was set to 1.0 for each experiment using untreated control cells. LC3II or p62 levels from all treated cells per experiment were normalized to this control. Next, rate of autophagic flux or p62 degradation, termed flux, was determined by dividing normalized LC3II or p62 in cells with N+L by LC3II or p62, respectively, in cells without N+L. Lastly, total amount of LC3II or p62 degradation, termed net flux, was determined by subtracting LC3II or p62 in cells without N+L from LC3II or p62 in cells with N+L [43,46]. The effect of HIV, ART, morphine, and morphine + ART was analyzed as a fold change relative to untreated uninfected or HIV-infected control cells. These general guidelines are used to interpret overall effects on in bulk autophagy: low steady-state LC3II levels and low LC3II flux would be indicative of decreased autophagy induction; high steady-state LC3II levels and high LC3II flux would be indicative of autophagy induction and proper maturation; high steady-state LC3II with reduced LC3II flux would be indicative of reduced maturation; high steady-state LC3II with no change in LC3II flux would be indicative of increased formation not met with increased maturation. To directly compare the impact of morphine and ART on autophagic flux and p62 degradation in uninfected vs. infected cells, we corrected the normalized protein values for infected cells to account for the presence of HIV using our uninfected and infected LC3II and p62 data. We first created conversion factors for values with and without N+L. Without N+L, this conversion factor was the average fold change in LC3II or p62 accumulation relative to the uninfected control, which was 1.25 for both data sets. For the N+L values, the conversion factor was calculated by multiplying the average LC3II or p62 value for N+L-treated infected cells by the average LC3II or p62 values for uninfected untreated cells, and this was divided by the average uninfected untreated LC3II or p62 value. For each individual experiment examining infected cells, all baseline values (no N+L) for untreated or treated cells were multiplied by the conversion factor, and for N+L values, the number was multiplied by the conversion factor and divided by the control N+L value within that experiment. This resulted in a new set of normalized LC3II or p62 values that were then compared to uninfected cell values. These numbers were also used to quantify relative flux or net flux by the same calculations mentioned above, and these were tested for significance per treatment between uninfected and infected cells. 

### 2.4. Immunofluorescence Microscopy

Uninfected or HIV-infected MDM were cultured on PDL-coated coverslips (Corning Inc., Corning, NY, USA) on 24-well plates as described and treated with morphine and/or ART for 24 h. N+L was added to one set of control or treated cells for the last 4 h. After treatment, cells were washed 3 times with 1 × PBS, fixed in 3.7% paraformaldehyde for 15 min at room temperature, and washed 3 times with 1x PBS. MDM were permeabilized for 2 min with 0.1% Triton-×-100 in PBS and blocked with a super blocking solution prepared freshly for each experiment. Each aliquot consisted of 9 mL ddH_2_O, 1 mL 0.5 M EDTA (Gibco Technologies, Amarillo, TX, USA), 100 μL 45% gelatin from cold-water fish (Sigma-Aldrich, St. Louis, MO, USA), 0.1 g immunoglobulin free bovine serum albumin (Sigma-Aldrich, St. Louis, MO, USA), 100 μL horse serum (Sigma-Aldrich, St. Louis, MO, USA), and 535 μL human AB serum (Corning Inc., Corning, NY, USA) [47]. After 30 min of blocking, coverslips were incubated at 4 °C overnight in primary antibodies against LC3B (520 ng/mL, 1:100 Cell Signaling Technology #2775, Danvers, MA, USA), p62 (2 μg/mL, 1:250 Enzo Life Sciences BML-PW9860, Ann Arbor, MI, USA), and/or TOM20 (2 μg/mL, 1:100 Santa Cruz Biotechnology F-10 sc-17764, Dallas, TX, USA) or a mouse/rabbit IgG isotype control (10 μg/mL, 1:100 or 25 μg/mL, 1:250 Invitrogen #02-6100/02-6102, Carlsbad, CA, USA) diluted in blocking solution. Coverslips were washed 3 times with 1 × PBS and incubated for 1 h at room temperature in secondary antibodies, goat anti-rabbit Alexa Fluor 488 (4 μg/mL, 1:500 Invitrogen #A-11008, Carlsbad, CA, USA), and/or goat anti-mouse Alexa Fluor 594 (4 μg/mL, 1:500 Invitrogen #A-11005, Carlsbad, CA, USA). After 3 washes in 1 × PBS, coverslips were mounted onto frosted microscope slides (Thermo Fisher Scientific, Waltham, MA, USA) with ProLong Diamond Antifade Mountant with DAPI (Invitrogen, Carlsbad, CA, USA) and cured at room temperature for 24 h. Slides were visualized in a blinded fashion at 40× in Z-series using a confocal Leica DMI8 microscope at the Analytical Imaging Facility at Albert Einstein College of Medicine. Z-series images were analyzed in Volocity ×64 (Quorum Technologies, Lewes, UK) to quantify the number of LC3 or p62 puncta, as well as total mitochondrial volume and colocalization of LC3 with TOM20. Signal and number of puncta were quantified in a blinded fashion in 40–80 cells per condition with distinct thresholds for puncta size and intensity, as well as TOM20 signal, set for each experiment individually. Mitophagosomes were quantified as LC3 puncta positive for mitochondria. Flux of LC3, p62, and mitophagosomes was calculated similarly to Western blotting experiments by dividing the number of puncta in cells with N+L by the number of puncta in cells without N+L per treatment followed by determining a fold change in treatment flux relative to control set to 1.0 in each experiment.

### 2.5. Real-Time Quantitative Polymerase Chain Reaction (RT-qPCR) 

Uninfected or HIV-infected MDM were cultured on 60 mm plates and left untreated or treated for 3, 6, or 24 h with morphine and/or ART. Plates were washed with cold 1 × PBS, and total RNA was extracted using Trizol according to the manufacturer’s protocol (Thermo Fisher Scientific, Waltham, MA, USA), including chloroform extraction, 2-propanol precipitation, and washing in 75% ethanol. Isolated RNA was diluted in RNase-free water (Ambion Inc., Austin, TX, USA) and quantified with a NanoDrop 2000 Spectrophotometer (Thermo Fisher Scientific, Waltham, MA, USA). A 2.0 μg amount of RNA per condition was reverse transcribed into cDNA using SuperScript Vilo Master Mix (Invitrogen, Carlsbad, CA, USA) according to the manufacturer’s protocol and stored at −20 °C prior to RT-qPCR. Taqman Gene Expression Assays (Applied Biosystems, Waltham, MA, USA) for *18S* reference gene (Cat. No.: Hs99999901_s1, Thermo Fisher Scientific, Waltham, MA, USA) or *p62/SQSTM1* gene (Cat. No.: Hs01061917_g1, Thermo Fisher Scientific, Waltham, MA, USA) were performed in Taqman Gene Expression Master Mix on a StepOne Plus Real-Time PCR system (Applied Biosystems, Waltham, MA, USA) using recommended conditions for Taqman Assays. Relative amount of *p62*/*SQSTM1* mRNA in morphine and/or ART-treated MDM was calculated using the 2^−ΔΔCt^ method relative to cDNA from control cells with *18S* as the reference gene. This resulted in a fold change in gene expression per treatment relative to control set to 1.0 for each experiment.

### 2.6. Statistical Analysis

All quantitative data were analyzed in Prism software v.8.0.1 (GraphPad Software Inc., San Diego, CA, USA). Data were tested for normality using the Shapiro–Wilk test with *p* = 0.05 as the cutoff. All data were normally distributed in which 2 or more groups were compared. These data were analyzed relative to control by the appropriate unpaired or paired Student’s *t*-test or by one-way ANOVA to analyze for significance when comparing more than 2 groups. When ANOVA was performed, this was followed by a multiple-comparison Dunnett’s test or a Turkey’s test to follow up differences between specific groups. For fold change analyses, one-sample *t*-tests were used for normally distributed data, and for data not normally distributed, Wilcoxon Signed Rank tests were used. For these analyses, values from untreated cells were set to 1.0. Values of *p* < 0.05 were considered statistically significant for all experiments.

## 3. Results

### 3.1. Morphine and ART Inhibit Autophagic Flux in the Context of HIV Infection

We first characterized the impact of morphine (100 nM) and ART (15 μM emtricitabine, 15 μM tenofovir, and 1 μM raltegravir) on autophagy in uninfected primary human monocyte-derived macrophages (MDM) by Western blotting. PBMC were isolated from leukopaks and cultured adherently into macrophages for 6 days with M-CSF prior to treatment with morphine and/or ART for 24 h. Lysates were collected and analyzed by Western blotting for LC3II. This form of LC3 is associated with the inner and outer membrane of forming autophagosomes (APG), making it possible to correlate LC3II steady-state levels with the number of APG present [28,31]. Autophagy is dynamic, and levels of LC3II reflect both autophagy induction and APG maturation. To examine both processes by Western blotting, lysosomal inhibitors are used to block LC3II degradation. We added 10–20 mM NH_4_Cl + 100–200 μM leupeptin (N+L), which neutralize the acidity of lysosomes and inhibit serine-cysteine lysosomal proteases, respectively, to cells in the last 4 h of 24 h treatment [44]. Levels of LC3II with N+L across treatments correlate with how many APG accumulate from autophagy induction until maturation. Although none of the treatments caused significant changes in steady-state LC3II levels in uninfected cells, we found a consistent upward trend with MOR+ART. Cells treated with MOR+ART displayed an increase in LC3II of 1.38-fold, which was mostly driven by ART, as ART treatment alone increased LC3II by 1.22-fold (Figure 1B, *p* = 0.08 one-sample *t*-test). An increase in steady-state LC3II can be a consequence of increased induction of autophagy (more formation of APG) or reduced clearance of formed APG. To discriminate between these two possibilities, we compared the changes in LC3II levels upon addition of N+L that allows calculating LC3II flux (ratio of LC3II+N/L to steady-state LC3II) as well as the total amount of LC3II degraded, net flux, defined as the subtraction of steady-state levels for LC3II from levels upon N+L addition. We found a trend with the combination of MOR+ART toward higher LC3II flux that becomes significant when considering the total amount of LC3II degraded under these conditions, net flux (Figure 1C,D). ART treatment alone did not increase autophagic flux, suggesting that the slight increase in steady-state levels of LC3II in cells treated with ART alone may be resultant from induction of APG formation that is not met by increased clearance (maturation). The increase in LC3II flux observed with morphine alone may be responsible for the higher autophagic flux with MOR+ART (Figure 1C,D). We propose that the persistence of elevated LC3II steady-state levels in cells treated with MOR+ART, despite the observed increase in flux, could reflect the inability of cells to accommodate fully for an ART-driven increase in APG biogenesis.

We next analyzed the impact of HIV infection on autophagy. Day 6 MDM were infected with HIV for 4–5 days, and Western blot analysis of LC3II was performed as described with N+L. Productive infection with HIV was confirmed in culture supernatants by HIV p24 alphaLISA (Perkin-Elmer, Waltham, MA, USA), and no p24 was detected in uninfected cell cultures (Figure 1E). Relative to no infection, HIV significantly increased steady-state levels of LC3II (Figure 1G) but did not significantly change LC3II flux or net flux (Figure 1H,I). Higher levels of LC3II without changes in overall flux can only be explained by a combination of increased APG biogenesis, autophagy induction, coincident with inhibited maturation of the newly formed APG. Impaired APG maturation with HIV was more evident when, instead of analyzing LC3II (which reflects flux through all types of macroautophagy), we analyzed levels and degradation of p62, a selective autophagy receptor for poly-ubiquitinated cargo [32]. P62 degradation by autophagy can be calculated similarly using N+L as described. It also accumulates when autophagic flux is defective, leading to a buildup of undegraded p62 and associated cargo. Overall, there was a trend toward increased baseline p62 and decreased lysosomal degradation of p62 in HIV-infected MDM compatible with a preferentially inhibitory effect of HIV infection on selective autophagy (Appendix A, *n* = 4).

Next, we analyzed possible synergistic effects on autophagy of morphine, ART, and HIV infection. When MDM were treated for 24 h after 3–4 days of infection, there was a trend toward increased p24 in culture supernatants with morphine alone, but it was not statistically significant (Figure 1E). This is consistent with published data indicating that morphine increases HIV infection of macrophages with increased infection time [48,49,50,51]. The impact of morphine ± ART on autophagy in HIV-infected MDM was more inhibitory than in uninfected MDM. Morphine ± ART significantly increased baseline LC3II levels relative to infected control cells (Figure 1K). In contrast to the increase in LC3II flux observed in uninfected cells treated with both drugs (Figure 1C,D), morphine with or without ART did not significantly change LC3II flux or net flux, and there was even a trend toward decreased flux with morphine alone (Figure 1L,M). We then compared our Western blotting data in uninfected MDM with data from infected MDM to quantify whether morphine and ART had different effects on autophagy in the context of HIV. To do this, we corrected the normalized LC3II levels in Figure 1K to account for the average presence of HIV and the impact of morphine and ART on uninfected cells from Figure 1B. Flux and net flux relative to control were recalculated. In infected MDM, morphine significantly increased baseline LC3II relative to uninfected cells treated with morphine resulting from significantly decreased flux in infected MDM (Figure 1N,O). Morphine + ART significantly reduced net flux in infected MDM compared to uninfected cells (Figure 1P) with trends toward increased LC3II levels (Figure 1N) and decreased flux (Figure 1O) as well. These data show that morphine, even in the presence of ART, also induces autophagy but further impairs APG maturation in HIV-infected cells relative to uninfected cells. We propose a synergistic effect of morphine and HIV on autophagy that leads to overall decreased autophagic flux.

### 3.2. Imaging Data Demonstrate Further That Morphine and ART Inhibit Autophagic Flux in HIV-Infected Macrophages

We complemented our biochemical analysis with immunofluorescence (IF) for LC3 to better characterize the proposed defect in APG maturation and determine whether the observed changes in LC3 levels were related to differences in the LC3 content per APG or in the overall number of APG. There was no significant staining nor visible puncta in the isotype-matched IgG control (Figure 2B). We analyzed LC3 puncta in Z-series by confocal IF in response to morphine and/or ART. By this technique, N+L prevents degradation of APG content upon fusion with lysosomes, which are visualized as more puncta (Figure 2A). We quantified in a blinded manner LC3 puncta per cell in Z-series with 40–80 cells per treatment. These values were averaged to produce a mean puncta/cell value per treatment in each experiment. Puncta values were used to quantify flux as a fold change relative to control set to 1.0 for each experiment. Like the Western blotting analysis, although results did not reach significance, uninfected cells treated with ART displayed a trend toward more puncta (Figure 2C, *p* = 0.1275 one-way ANOVA) and decreased flux (Figure 2D, *p* = 0.059 one-sample *t*-test), suggesting potential impairment of APG maturation. Interestingly, our results with morphine ± ART were different from our Western blotting analysis. This may be due to variation in the amount of LC3II present on APG from different donors as measured by Western blotting, variation between donors in the raw optical density values for LC3II, and/or morphine having different effects on distinct APG subgroups (with different overall LC3 content), with some still maturing properly and others failing to mature. This last possibility is supported by the fact that, in the presence of morphine, content inside APG that fuses with lysosomes is degraded more readily (Figure 1), but the number of APG that fuse with AL in the presence of morphine appears to be smaller relative to control (Figure 2D). In contrast, morphine + ART did not change any of these values significantly compared to untreated cells, supporting that these drugs affect degradation of APG differently when considering the number of organelles or overall LC3 content per organelle. These data indicate that short-term treatments for 24 h with morphine and ART may have subtle but consistent inhibitory effects on maturation of at least a subgroup of APG in uninfected MDM.

We also quantified LC3 puncta by confocal IF with morphine and ART in HIV-infected MDM. The number of quantifiable puncta again increased as expected with N+L due to defective maturation of APG into AL (Figure 2E). Without N+L, there was a trend toward increased LC3 puncta with morphine ± ART, but these were not statistically significant. With N+L, cells treated with morphine showed almost no increase in the number of LC3 puncta, and there was a trend toward a smaller increase in the number of puncta with morphine + ART (Figure 2F). Using these values with and without N+L, we calculated overall LC3 flux. Morphine ± ART significantly decreased LC3 flux (Figure 2G). These microscopy-based assays demonstrated that morphine with or without ART significantly impairs APG formation and maturation in HIV-infected MDM resulting in decreased autophagic flux. This underscores the significant inhibitory effect that morphine treatment has on autophagy in human macrophages in the context of HIV, even in the presence of several antiretroviral drugs. 

### 3.3. Morphine and ART Preferentially Inhibit p62-Mediated Selective Autophagy

To determine the possibility of different effects of HIV, morphine, and ART on distinct types of APG, we next analyzed their impact on selective forms of autophagy. One of the most well-characterized receptors for selective autophagy is p62 [52]. Once p62 is inside APG, it is degraded in autolysosomes (AL). Lysosomal flux of p62 can be measured similarly to LC3 by Western blotting and IF. However, levels of detectable p62 degradation at baseline vary greatly from total LC3 flux in primary human macrophages due to cargo selectivity. While N+L caused significant accumulation of LC3II in MDM from every individual donor by Western blotting, this did not occur for p62. When N+L caused accumulation of p62 at least 1.2-fold more than in control cells, this was considered detectable lysosomal degradation of p62. Of the MDM infected in vitro from 11 donors for which we measured LC3II flux by Western blotting, 7 had detectable p62 degradation (Figure 1K and Figure 3F). Rate of p62 degradation (flux) and total amount of p62 degraded (net flux) were calculated similarly to LC3II. 

By Western blotting, morphine and/or ART increased p62 levels and decreased autophagic degradation of p62 regardless of infection. This provides further evidence that morphine and ART preferentially impact certain types of autophagy both with and without HIV. In uninfected MDM, morphine, or ART alone, significantly increased p62 levels. In the presence of N+L, there was no change (Figure 3B). This corresponded to a significant decrease in p62 degradation with morphine or ART and a significant decrease in the overall amount of p62 degraded in lysosomes with ART alone (Figure 3C,D). This impact of morphine with or without ART on p62 levels and degradation persisted in HIV-infected MDM. Morphine ± ART significantly increased steady-state levels of p62 (Figure 3F), and both treatments significantly decreased the rate and amount of p62 degradation by autophagy (Figure 3G,H). These decreases were more consistent than in uninfected cells treated with morphine ± ART. Like our LC3II analysis, we corrected normalized p62 values in infected MDM to account for the average presence of HIV, as in Appendix A, and the impact of morphine and ART on p62 in uninfected cells as in Figure 3B. We did not detect any significant differences in p62 levels or its lysosomal degradation between uninfected and infected cells treated with morphine in the presence or absence of ART (Figure 3I–K). This is likely because morphine and ART preferentially impair selective p62-mediated autophagy regardless of HIV infection status.

Unlike LC3, total p62 levels are also regulated highly by transcriptional programs [53,54]. To confirm that changes in p62 levels were directly mediated autophagy and unrelated to transcription, we measured *p62/**SQSTM1* mRNA levels by RT-qPCR. In uninfected MDM, we treated MDM with morphine and/or ART for 6 h and 24 h and observed no changes relative to control (Figure 4A,B).

Transcriptional upregulation of *p62/SQSTM1* can also occur at earlier time points, so in HIV-infected MDM, we measured mRNA after 3 h and 6 h of treatment [54]. Like in uninfected MDM, there was no change in transcription relative to infected controls (Figure 4C,D). Thus, the observed changes in p62 protein levels are likely due to their reduced degradation by autophagy and not related to transcription. To determine whether changes in p62 degradation by autophagy were due to reduced APG formation/sequestration of p62 or a consequence of the proposed reduced maturation in APG demonstrated by our LC3II flux studies, we analyzed by confocal IF the number of p62 puncta (likely p62 already in APG) in the presence and absence of N+L in response to morphine ± ART (Figure 5A). There was a trend toward more p62 puncta at baseline with morphine ± ART relative to untreated cells, corresponding to a significant decrease in lysosomal degradation of p62 (Figure 5B,C). These data confirmed our results by Western blotting and support that, rather than problems in APG biogenesis, higher p62 levels, and reduced degradation were consequences of reduced APG maturation. This may suggest decreased degradation by autophagy of polyubiquitinated protein aggregates that can further dysregulate cell function and homeostasis to contribute to neuropathogenesis.

### 3.4. Morphine with or without ART Impairs Mitophagic Flux in HIV-Infected MDM

Selective degradation of damaged mitochondria by macroautophagy, termed mitophagy, is important for maintaining cellular quality control [55]. Disruption of the electron transport chain by various toxins induces mitophagy to eliminate faulty mitochondria so that functional organelles can be regenerated [56]. Failure to complete mitophagy causes buildup of defective mitochondria. We showed that morphine and ART impair selective autophagic degradation of p62; therefore, we examined whether other selective autophagic processes, such as mitophagy, were affected in HIV-infected MDM. We performed confocal IF colocalization studies in Z-series of LC3 (green) with TOM20 (red), a mitochondrial outer membrane protein used commonly to monitor mitophagy by colocalization of the mitochondria and APG proteins (Figure 6A) [57]. There was a consistent trend toward increased total mitochondrial volume per cell with N+L relative to control, indicating detectable mitophagic flux (Figure 6B,C). Furthermore, ~10–15% of total APG/cell by LC3 staining were positive for TOM20 at baseline with an average of ~45 mitophagosomes/cell per Z-series. This increased reliably with N+L due to expected buildup of APG with undegraded mitochondria (Figure 6D). Morphine + ART significantly increased the number of mitophagosomes per cell, and morphine alone appeared to increase this as well (Figure 6D). There was no significant change in mitophagosomes in the presence of N+L, which is consistent with our LC3II Western blot data and suggests that morphine inhibits degradation of mitochondria by autophagy in HIV-infected MDM (Figure 6D). Using these numbers, we calculated overall mitophagic flux, which decreased significantly with morphine and appeared to decrease with morphine + ART as well (Figure 6E). This matched our IF results for LC3 and p62 and confirmed that morphine and ART appear to inhibit this form of selective autophagy. Morphine alone also significantly increased the % of APG positive for mitochondria, suggesting concomitant induction of mitophagy that also corresponds with our LC3II Western blot data in Figure 1. This trend was similar with morphine and ART together (Figure 6F). Thus, the impact of morphine and ART on mitophagy in HIV-infected MDM corresponds with changes in total autophagy by LC3II Western blotting. These changes may cause accumulation of defective mitochondria inside APG that cannot properly regulate cell homeostasis.

## 4. Discussion

Autophagy is a dynamic quality control process essential for maintaining homeostasis in long-lived cell types such as macrophages. Basal autophagic flux maintains vital functions of the cell ranging from metabolism to cell survival by degrading damaged and faulty organelles and macromolecules [28]. Various studies have shown that autophagy is dysregulated in neurodegeneration, including in models of Alzheimer’s, Parkinson’s, and Huntington’s [58,59,60,61,62,63]. A few studies have also linked dysregulated autophagy to HAND pathogenesis and to peripheral control of HIV viral loads [64,65,66]. Overall, increased autophagic activity appears to have beneficial effects, although excessive autophagy induction can also be toxic [67]. No other studies, to our knowledge, have addressed the impact of opioids and commonly prescribed ART drugs on autophagy in uninfected and HIV-infected macrophages, which are reservoirs for HIV in the CNS. It is important to characterize these effects to develop strategies of therapeutic manipulation of autophagy that potentially can treat people with HAND using opioids. In this work, we show a previously unknown interaction of opioids, ART drugs, and HIV on autophagy that could help explain the accelerated deterioration of this process in ART-suppressed individuals that may contribute to higher incidence of HAND (Figure 7).

Our results indicate that 24 h of morphine at concentrations equivalent to average CNS levels in people using opioids has mixed effects on autophagy in primary human monocyte-derived macrophages (MDM) [68,69,70]. Using primary cells highlights the inherent variability among people in response to morphine. Our Western blotting data indicate that morphine induces autophagy, even in the presence of ART, leading to overall increased flux (Figure 1C). Morphine-mediated autophagy induction is supported by additional Western blot data demonstrating that morphine significantly increases ATG5 and ATG7 in uninfected cells (Appendix A). These two proteins are essential for early steps of autophagy activation, but levels of these factors do not always correlate with levels of autophagy induction [43]. Increased factors that participate in induction may mediate the ability of uninfected MDM to upregulate flux more efficiently than infected cells when exposed to morphine. We hypothesized that increased ATG5 and ATG7 may be due to stimulation of the Transcription Factor EB (TFEB) pathway, which is activated after mTOR inhibition to express proteins necessary for autophagy and for lysosomal biogenesis [71]. To study this, we examined levels of *TFEB* mRNA at 3 h and 6 h of morphine with/without ART, and we found no changes relative to control (Appendix A). It is possible that the TFEB pathway is not induced until later time points, however, or that other mechanisms are inducing autophagy downstream of the μ-opioid receptor [72]. One study in a neuroblastoma cell line demonstrated that morphine induces autophagy by dissociating BECLIN-1, part of the class III PI3K complex needed to produce PI(3)P necessary for autophagosome (APG) formation, from B-cell Lymphoma-2 (BCL-2) [73]. BECLIN-1 also has redundant roles in APG maturation, however, making it difficult to draw conclusions about its role in autophagy manipulations in general [33,74]. We did not find any changes in BECLIN-1 by Western blotting with morphine and/or or ART (Appendix A). However, if induction of autophagy by morphine is independent of TFEB activation, and hence will not be associated with the TFEB-dependent increase in lysosomal biogenesis, then the number of these organelles available for APG clearance could become limiting long-term. This would predispose cells to excessive accumulation of APG upon additional autophagy induction over time.

The apparent discrepancy between our LC3 Western blotting and IF results in uninfected MDM highlights the importance of using multiple approaches to study autophagy, as both assays analyze different autophagy properties. Western blotting informs overall cellular content, while IF provides information about the number of autophagic vesicles and, consequently, changes in content per vesicle [43]. In fact, only through IF were we able to detect that morphine also blocks APG maturation (Figure 2). This dual effect on autophagy is consistent with one study in bone-marrow-derived macrophages studying the effects of lipopolysaccharide (LPS) in the context of morphine [75]. LPS is increased systemically in PWH due to bacterial translocation from the gut early on in HIV infection [76,77]. This study showed that 1 μM morphine pretreatment for 24 h prior to LPS activation for 24 h increased LC3 accumulation due to both induced autophagy and inhibited APG maturation. Inhibited APG maturation was caused by defective acidification of the lysosomes, which impairs autolysosome (AL) formation and reduces flux [75]. This inhibitory effect on lysosome function in response to morphine tested at 500 nM was recapitulated in human microglia [78]. We assayed intracellular pH by a fluorometric plate reader method in response to morphine and ART in HIV-infected MDM and saw no significant changes (Appendix A). We also performed Western blots for Lysosome Associated Membrane Protein 1 (LAMP1), an endolysosome marker, and mature lysosomal proteases, such as Cathepsin D, in uninfected MDM, and saw no significant changes relative to control (Appendix A). No change in these proteins suggests that lysosomal biogenesis may not be upregulated to accommodate for the increase in APG biogenesis with morphine and ART. These data could also indicate that impaired maturation in our system, which tested morphine concentrations 5–10x lower than in previously mentioned studies, may occur through mechanisms unrelated to lysosome quantity and function.

The same inhibitory impact on lysosome function and APG maturation was found in primary rat microglia exposed to a similar ART cocktail of tenofovir, emtricitabine, and dolutegravir [79]. Although our tested ART cocktail containing raltegravir instead of dolutegravir did not produce statistically significant changes in total autophagy in uninfected MDM, the trends were consistent with impaired maturation reducing flux (Figure 1 and Figure 2). By Western blotting, the flux response to ART alone in uninfected cells also appeared clustered, with approximately half showing an increase and half showing a decrease (Figure 1C,D). In HIV-infected MDM, ART alone had negligible effects on both steady-state levels of LC3II and flux or net flux by Western blotting (Appendix A). There were also no significant differences in any observed autophagy parameters between HIV-infected MDM treated with morphine and HIV-infected MDM treated with morphine and ART. These data together demonstrate that inhibited autophagy in the context of HIV, opioids, and ART is mediated likely by morphine alone.

HIV appears to increase the inhibitory effects on APG maturation of morphine in human macrophages, as shown graphically in Figure 7. This is likely related to how HIV itself manipulates autophagy. A few studies showed that HIV induces autophagy in human macrophages, and one found that HIV impacts autophagy differentially during induction and maturation using cell lines [34,35]. While autophagy appears to be induced early on to augment virus production in forming APG, HIV Nef sequesters TFEB in the cytoplasm. TFEB sequestration blocks APG maturation to prevent viral particles from degradation in AL [35]. This causes net zero or decreased flux relative to uninfected cells, and cargo inside APG cannot be degraded effectively, including viral particles. We recapitulated this dual impact on autophagy in HIV-infected macrophages and showed that LC3II accumulates in infected MDM without any significant changes in flux or net flux (Figure 1G–I). Compromise of the machinery needed for maturation by HIV may render macrophages more susceptible to additional autophagic stress caused by morphine. Thus, flux cannot be upregulated as efficiently as in uninfected cells, which could have deleterious functional consequences. 

Inability of infected MDM to upregulate flux may also be due to the type of cellular stress that morphine causes to induce autophagy. Several stimuli, including oxidative stress, ER stress, and lipid-induced stress, induce a biphasic effect on autophagy, particularly on selective autophagy, with upregulation of autophagy to neutralize stress that often results in autophagy inhibition in cases of chronic persistence of the stressor [29]. Infected MDM may not mitigate the specific type of stress caused by morphine ± ART as well as uninfected cells, promoting autophagy inhibition. Our data in both uninfected and infected MDM indicate that morphine and ART impair selective autophagy mediated by p62 (Figure 3 and Figure 5). This cargo receptor docks polyubiquitinated cargo, linked most often at K63, for autophagic degradation in a selective process, aggrephagy. Morphine and ART inhibited p62 lysosomal degradation regardless of HIV infection. These results suggest that the pool of APG carrying aggregated proteins docked by p62 in primary human macrophages may be exquisitely sensitive to blocked APG maturation in response to morphine and ART. Inhibited p62 degradation should lead to a buildup of polyubiquitinated proteins that cannot properly be degraded, which could further impair cell functions to contribute to CNS macrophage dysregulation and HAND. Future studies will determine whether morphine and ART cause a buildup of these dysfunctional proteins. These cargo proteins may play important roles in maintaining homeostatic functions or altering inflammatory properties. 

Another well-characterized form of selective autophagy is mitophagy, which clears damaged/dysfunctional mitochondria so that functional organelles can be synthesized de novo [55]. Oxidative stress is a classic trigger of mitophagy [80]. There are distinct signaling programs through which mitophagy can occur, with the PINK1/Parkin pathway being the most understood [81]. As a result of our findings that in bulk autophagy and p62-mediated selective autophagy were impaired with morphine treatment of infected MDM, we determined whether mitophagy was also impacted. We found that the effects on mitophagy appeared to mirror effects on total autophagy. Our studies of the percentage of LC3 puncta positive for mitochondria by TOM20 staining showed an increase with morphine ± ART, which suggests accumulation of mitophagosomes (Figure 6F). When we quantified the number of mitophagosomes, we found that, in the presence of morphine with/without ART, mitophagosome clearance was significantly reduced, which suggests impaired autolysosomal degradation of mitochondria (Figure 6D,E). Overall, manipulations in mitophagy we detected in this study are suggestive of altered mitochondrial function. Future studies will determine whether morphine preferentially alters mitophagy or if changes in mitophagy occur due solely to impacted in bulk autophagy.

Changes in autophagy have been shown to alter functions of macrophages that are important for proper CNS health, including phagocytosis, ROS secretion/mitigation, and inflammatory cytokine release. When dysregulated chronically, these functions contribute to development of HAND, which can worsen with opioid use [10,11]. One group genetically ablated autophagy in murine macrophages and found that uptake of *M. tuberculosis* by phagocytosis increased without proper intracellular clearance [82]. This led to higher bacterial burdens systemically. Inhibited autophagy caused accumulation of p62, which through the Nuclear Factor Erythroid 2-Related Factor 2 (NRF2)/ Kelch-like ECH-associated Protein 1 (KEAP1) pathway, transcriptionally upregulated macrophage scavenger receptors that increased phagocytic uptake [82]. Conversely, another study in bone-marrow-derived murine macrophages found that phagocytosis of latex beads or apoptotic thymocytes decreased when ATG7 was knocked out [83]. Autophagy is triggered in response to ROS to mitigate negative impacts on metabolism and cell survival [84]. Lack of autophagy was shown to increase ROS levels and diminish mitochondrial fitness in murine macrophages, particularly in cells from older animals [83]. NLRP3 inflammasome-mediated rapid release of IL-1β, a potent and neurotoxic inflammatory cytokine, is also regulated by autophagy [85]. These inflammasome complexes are shuttled into forming APG, where they are degraded after maturation. Inhibited autophagic activity has been shown to increase IL-1β secretion due to lack of inflammasome degradation [86,87]. More studies are needed to confirm whether the changes in total and selective autophagy we observed in response to morphine and ART in HIV-infected cells alter these functions of CNS macrophages to increase neuropathogenesis in people with HAND who use opioids. Our studies underscore the importance and significance of developing novel therapies for these individuals that increase macroautophagy activity to restore appropriate macrophage functions that are critical to CNS homeostasis.

## Figures and Tables

**Figure 1 cells-10-02183-f001:**
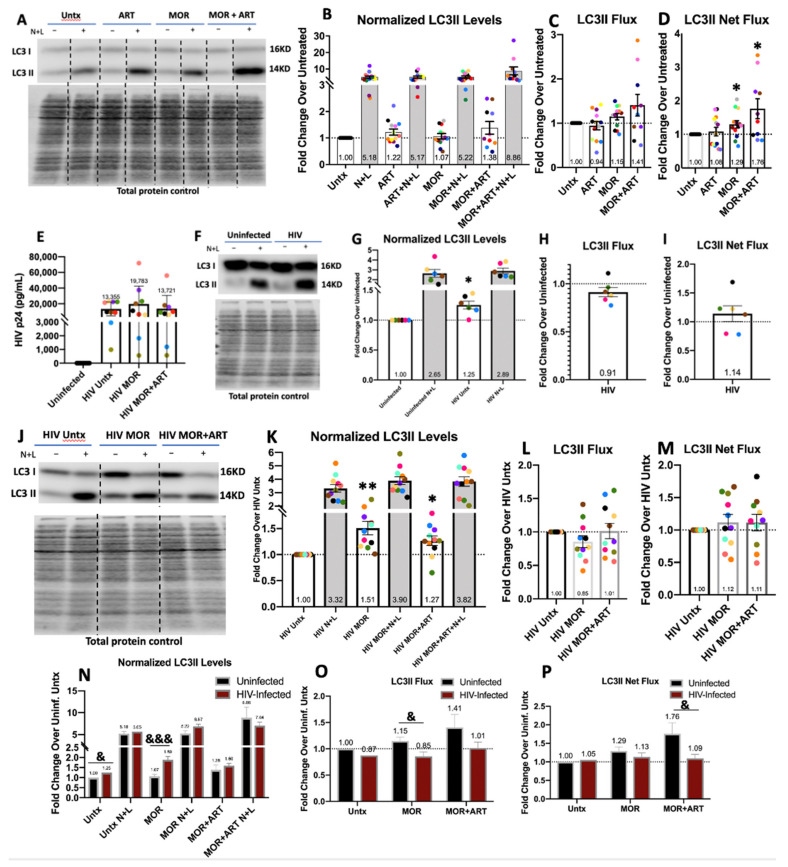
Normalized LC3II levels, LC3II flux, and LC3II net flux analyzed by Western blotting in uninfected and HIV-infected primary human MDM. Primary human MDM were cultured from PBMC and either left untreated (Untx) or treated with morphine (MOR: 100 nM) and/or antiretroviral therapy (ART: 15 μM tenofovir, 15 μM emtricitabine, 1 μM raltegravir) for 24 h with lysosomal inhibitors, NH4Cl and leupeptin (N+L), added in the last 4 h. Other cells were infected with HIV for 3-4 days and treated with MOR and/or ART for 24 h with N+L added in the last 4h. (**A**) Representative Western blot with total protein loading control in uninfected cells. (**B**) LC3II levels relative to total protein and normalized to the untreated control for each experiment. Each colored dot represents cells from a different person, and these colors are consistent across all data with uninfected MDM. (**C**) LC3II flux, equivalent to the rate of autophagic activity per treatment condition relative to control set to 1.0 at the dotted line, was quantified. (**D**) LC3II net flux, equivalent to the amount of total autophagic activity per treatment condition relative to control set to 1.0 at the dotted line, was quantified. (**E**) HIV p24 concentrations in culture supernatants for each treatment condition were determined by p24 alphaLISA. (**F**) Representative Western blot for untreated uninfected MDM vs. HIV-infected MDM with total protein loading control. (**G**) LC3II levels relative to total protein and then normalized to the untreated control for each experiment. Each colored dot represents cells from a different donor, and these colors are consistent across all data with HIV-infected MDM. (**H**) LC3II flux was quantified. (**I**) LC3II net flux was quantified. (**J**) Representative Western blot for untreated and treated HIV-infected MDM with total protein loading control. (**K**) LC3II levels relative to total protein and then normalized to the untreated HIV-infected control for each experiment. Each colored dot represents cells from a different person, and these colors are consistent across all data with HIV-infected MDM. (**L**) LC3II flux was quantified. (**M**) LC3II net flux was quantified. (**N**) Normalized LC3II values for infected MDM were corrected for the presence of HIV on average, and comparisons were made between MOR and MOR+ART treatments in uninfected and infected cells. (**O**) LC3II Flux values for infected MDM were corrected using the Normalized LC3II values in (**N**) and compared to uninfected flux values. (**P**) LC3II Net Flux values for infected MDM were corrected using the Normalized LC3II values in (**N**) and compared to uninfected flux values. Error bars depict SD for 1E and SEM for all other panels, *n* = 6–13 independent experiments, * *p* < 0.05, ** *p* < 0.01 one-sample *t*-test, & *p* < 0.05, &&& *p* < 0.001 unpaired Student’s *t*-test.

**Figure 2 cells-10-02183-f002:**
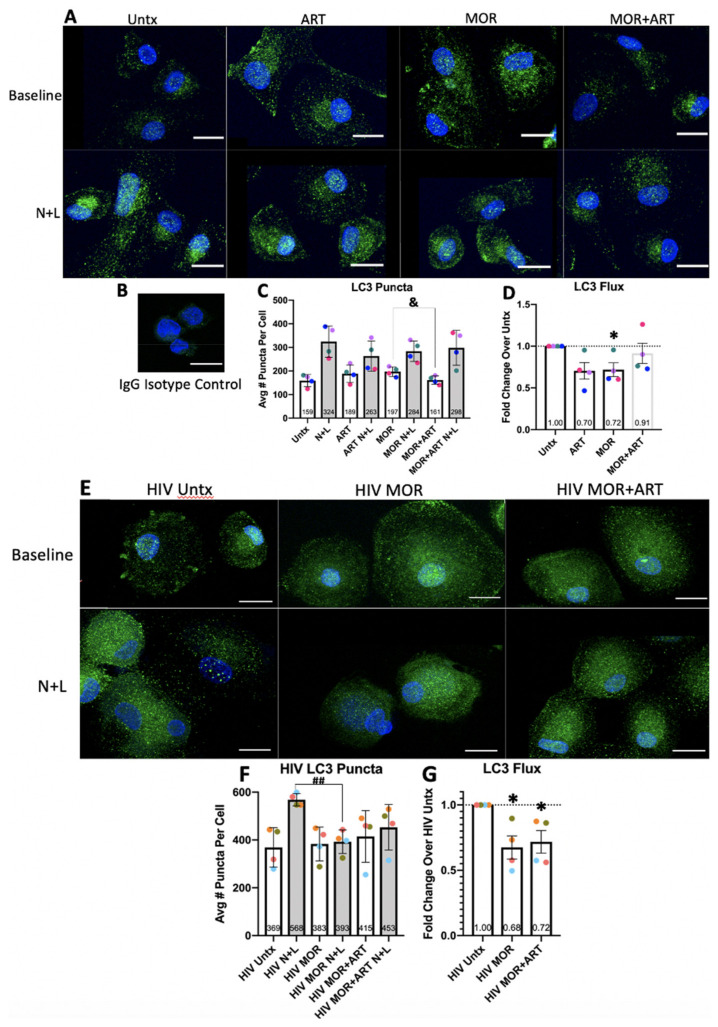
LC3 immunofluorescence studies in uninfected and HIV-infected MDM. Primary human macrophages were cultured on coverslips, infected with HIV or not, left untreated (Untx or HIV Untx) or treated with morphine and/or ART for 24 h with N+L added to some cells in the last 4 h of treatment, and coverslips were stained for LC3. Cells were imaged by confocal microscopy in Z-series, and LC3 puncta/cell were determined. (**A**) Representative uninfected untreated cells or cells treated with morphine and/or ART in the presence and absence of N+L. (**B**) Image of an IgG isotype-matched negative control at equivalent confocal laser exposure. (**C**) Average number of LC3 puncta per cell across treatments. (**D**) LC3 flux calculated relative to untreated control set to 1.0 using values for average number of LC3 puncta/cell. (**E**) Representative HIV-infected untreated cells or cells treated with morphine and/or ART in the presence and absence of N+L. (**F**) Average number of LC3 puncta per cell across treatments. (**G**) LC3 flux calculated relative to untreated control set to 1.0. Scale bar is 15 μm. Puncta/cell were quantified in a blinded fashion in 40–80 cells per treatment condition for each experiment and averaged. Error bars for puncta values represent SD, and error bars for LC3 flux represent SEM, *n* = 4 independent experiments, * *p* < 0.05, one-sample *t*-test, & *p* < 0.05 one-way ANOVA MOR compared to MOR+ART, ## *p* < 0.01, one-way ANOVA HIV MOR N+L compared to HIV N+L.

**Figure 3 cells-10-02183-f003:**
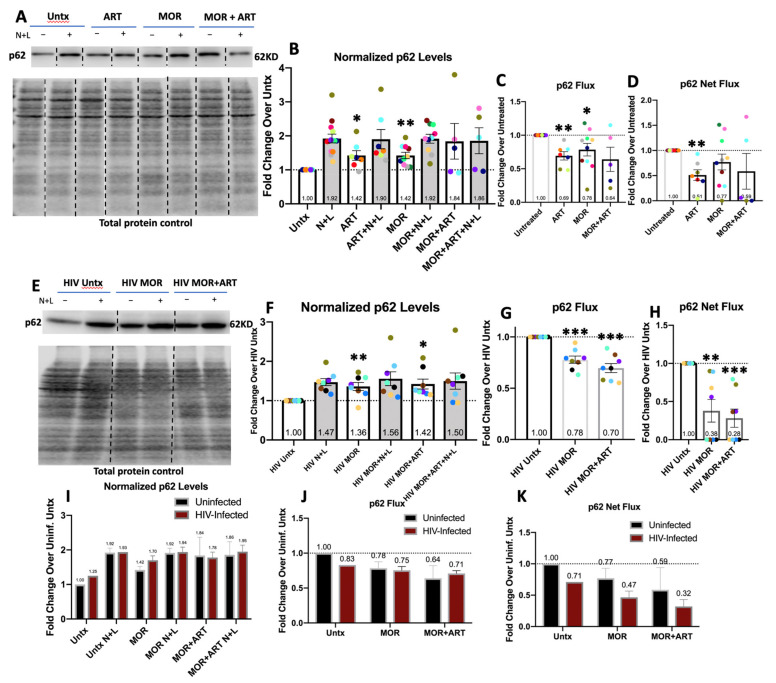
Normalized p62 levels, p62 flux, and p62 net flux analyzed by Western blotting in uninfected and HIV-infected macrophages. Primary human MDM were cultured as in the Materials and Methods section (Section 2) and left untreated (Untx) or treated with morphine and/or ART for 24 h with lysosomal inhibitors, NH4Cl and leupeptin (N+L) added in the last 4 h. (**A**) Representative Western blot with total protein loading control in uninfected cells. (**B**) p62 levels relative to total protein and normalized to the uninfected untreated control for each experiment. (**C**) p62 flux was quantified. (**D**) p62 net flux was quantified. (**E**) Representative Western blot with total protein loading control in HIV-infected cells. (**F**) p62 levels relative to total protein and normalized to the HIV-infected untreated control for each experiment. (**G**) p62 flux was quantified. (**H**) p62 net flux was quantified. (**I**) Normalized p62 values for infected MDM were corrected for the presence of HIV on average based on data in Appendix A according to calculations described in the Materials and Methods section (Section 2), and comparisons were made between MOR and MOR+ART treatments in uninfected and infected cells. (**J**) p62 Flux values for infected MDM were corrected using the Normalized p62 values in (**I**) and compared to uninfected flux values. (**K**) p62 Net Flux values for infected MDM were corrected using the Normalized p62 values in (**I**) and compared to uninfected flux values. Error bars depict SEM, *n* = 5–10 independent experiments, * *p* < 0.05, ** *p* <0.01 one-sample *t*-test, *** *p* < 0.001 one-sample *t*-test.

**Figure 4 cells-10-02183-f004:**
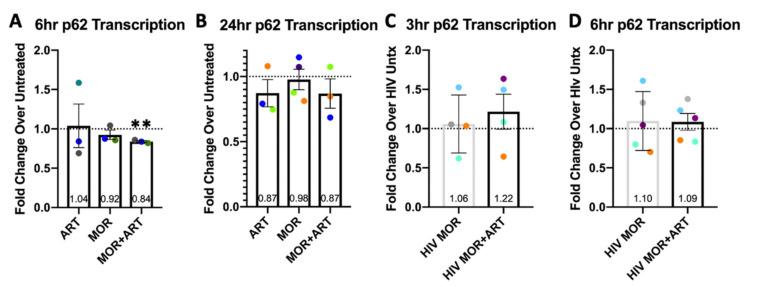
Measurements of *p62/SQSTM1* transcription in uninfected and HIV-infected primary human macrophages. MDM were cultured, infected with HIV or not for 3–4 days, and left untreated (Untx) or treated with morphine and/or ART for 3, 6, or 24 h as indicated. RNA was isolated, reverse transcribed to cDNA, and RT-qPCR analysis was performed with 18S as the reference gene. (**A**) *p62/SQSTM1* transcription in uninfected MDM at 6 h of treatment relative to control by 2^−∆∆Ct^. (**B**) *p62/SQSTM1* transcription in uninfected MDM at 24 h of treatment relative to control. (**C**) *p62/SQSTM1* transcription in HIV-infected MDM at 3 h of treatment relative to infected control. (**D**) *p62/SQSTM1* transcription in HIV-infected MDM at 6 h of treatment relative to infected control. *n* = 3–5 independent experiments with error bars representing SEM, ** *p* < 0.01 one-sample *t*-test.

**Figure 5 cells-10-02183-f005:**
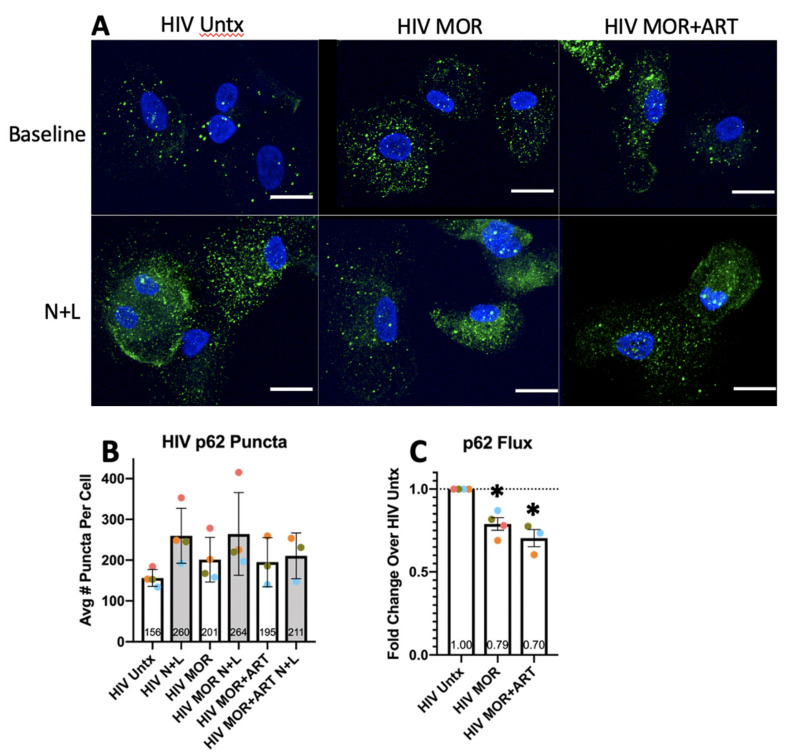
p62 immunofluorescence studies in HIV-infected MDM. Primary human macrophages were cultured on coverslips, infected with HIV for 3–4 days, and left untreated (HIV Untx) or treated with morphine and/or ART for 24 h with N+L added to some cells in the last 4 h of treatment. Coverslips were stained for p62 and imaged by confocal microscopy in Z-series, and p62 puncta/cell were determined with size and intensity thresholds set individually for each experiment. (**A**) Representative infected untreated cells or cells treated with morphine and/or ART with/without N+L. (**B**) Average number of p62 puncta per cell across treatments. (**C**) p62 flux was calculated relative to untreated control set to 1.0. Scale bar is 15 μm. Puncta/cell were quantified in a blinded manner in 40–80 cells per treatment condition for each experiment and averaged. Error bars for puncta values represent SD, and error bars for p62 flux represent SEM, *n* = 3–4 independent experiments, * *p* < 0.05 one-sample *t*-test.

**Figure 6 cells-10-02183-f006:**
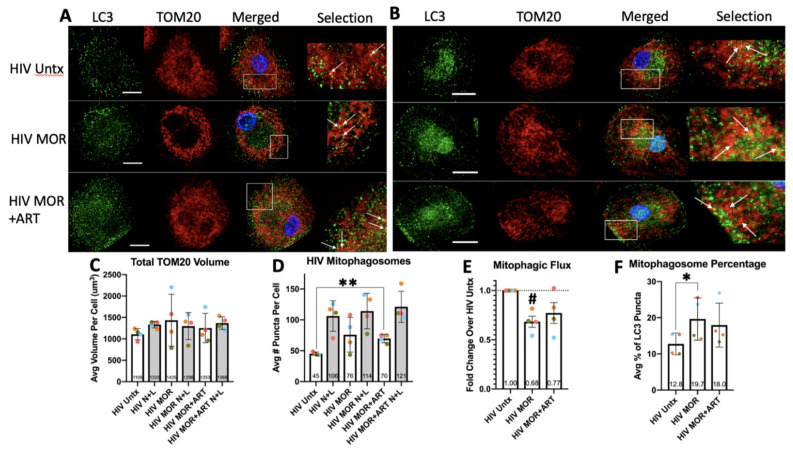
Analysis of mitophagy in HIV-infected MDM in response to morphine with/without ART by immunofluorescence. HIV-infected macrophages were cultured on coverslips and stained for LC3 (green), representing autophagosomes, and TOM20 (red), a mitochondrial protein representing mitochondrial mass. Coverslips were imaged in Z-series by confocal microscopy. (**A**) Representative images of infected MDM that were untreated (HIV Untx) or treated with morphine with/without ART. Arrows point to mitophagosomes, which are LC3 puncta positive for mitochondria. (**B**) Representative images of infected MDM treated or not and incubated with N+L in the last 4 h of treatment. Arrows point to mitophagosomes, LC3 puncta positive for mitochondria. (**C**) Total volume of TOM20 staining per cell was quantified per treatment. (**D**) Average number of mitophagosomes per cell was quantified per treatment. (**E**) Using the average number of mitophagosomes represented in (**D**) per experiment, flux was calculated relative to the infected untreated control set to 1.0. (**F**) The percentage of total LC3 puncta per cell positive for mitochondria was calculated and averaged per treatment in each experiment. Scale bar is 15 μm. Puncta or volume/cell was quantified blindly in 40–80 cells per treatment condition for each experiment and averaged. Error bars for puncta values, TOM20 volume, and % mitophagosomes represent SD, and error bars for mitophagic flux represent SEM, *n* = 4 independent experiments, * *p* < 0.05, ** *p* < 0.01 by one-way ANOVA, # *p* < 0.05 one-sample *t*-test.

**Figure 7 cells-10-02183-f007:**
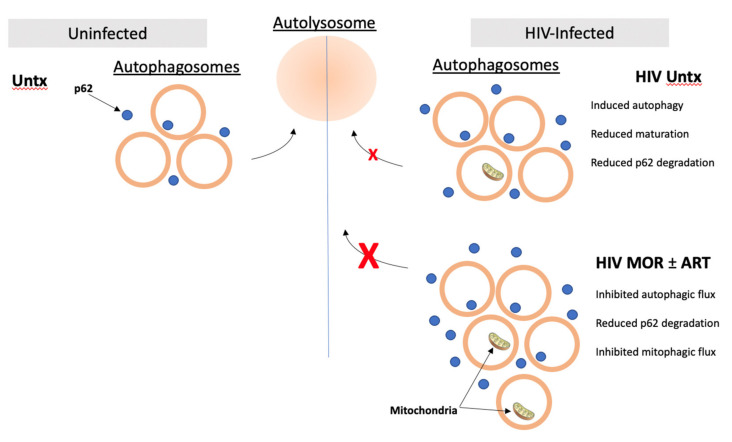
Schematic summarizing the effects of morphine (MOR) and ART for 24 h on in bulk autophagy, p62-mediated autophagy, and mitophagy in the context of HIV from autophagy induction to autophagosome (APG) maturation into autolysosomes (curved arrows). HIV infection of primary human macrophages (HIV Untx) increases the number of APG present by inducing autophagy and inhibiting maturation. HIV increases p62 levels by inhibiting selective autophagy. MOR+ART treatment of HIV-infected macrophages further induces autophagy and inhibits maturation, increasing the number of APG in infected cells. MOR+ART treatment also inhibits p62 flux and mitophagy in HIV-infected MDM, increasing p62 levels and the number of APG containing undigested mitochondria. Inhibited autophagy in macrophages, which are CNS reservoirs for HIV, may increase neuropathogenesis in people with HIV taking opioids and ART.

## Data Availability

Not applicable.

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
