# Peer review of "HIV Increases the Inhibitory Impact of Morphine and Antiretrovirals on Autophagy in Primary Human Macrophages: Contributions to Neuropathogenesis"

_cells, 2021, doi:10.3390/cells10092183_

Round 1
Reviewer 1 Report
Hi
The current manuscript is well developed and explained. Only minor suggestions
- line 22 and 23. the sentence can be split into 2 for clarity
- Use the Greek signs for micro and other abbreviations
- Add information about the salt used for each ART. Line 125
- degrees or oC
- there are several omissions of spaces and missing letters
- For Fig. 2. Are these pictures 3D reconstructions? clarify
- in Fig. 6 staining for TOM 20 is too strong. Please reduced to avoid problems of oversaturation
Reviewer 2 Report
Summary: The manuscript presents studies that indicate morphine and antiretroviral drugs treatments in the context of HIV infection can adversely affect macrophage autophagy. This is manifested as an induction in bulk autophagy accompanied by inhibition of its completion. Selective autophagy mediated by p62 and mitophagic flux were also affected. These alterations in macrophage autophagy might play a role in HIV-associated neurocognitive dysfunction in people using opioids.
Overall Comments: This is a well-written paper that presents strong evidence for altered autophagy in macrophages upon exposure to morphine (opioids) and a standard antiretroviral therapy cocktail with and without HIV infection.
Specific Comments:
- Provide a justification for the concentrations of ART used for cell treatment. Were the tenofovir, emtricitabine and raltegravir the free base drugs or in salt form? Was tenofovir used as disoproxil fumarate prodrug or the tenofovir itself?
- What is the percentage/purity of macrophages in the cultures derived from PBMCs? Provide a reference if available.
- Figure 1C, D, H ,I, L, M, and P are not accessible in the pdf provided for review. Thus, these panels could not be
- While the authors gave a justification for not including an HIV-infected ART-treated control, the inclusion of this control would provide needed context for the interpretations of MOR effects in conjunction with ART.
- In Figures 1, 2, 3, 5 provide a definition for the abbreviation “Untx”.
- Figure 6A and B and Figure 7 are not referred to in the text.
- The legend for Figure 7 needs to be more descriptive of what is being illustrated in the figure.
- The discussion contains 6 references to data not shown. This data should be included in the supplemental data.
Round 2
Reviewer 2 Report
The authors revisions adequately address the review comments.